# Leukotrienes vs. Montelukast—Activity, Metabolism, and Toxicity Hints for Repurposing

**DOI:** 10.3390/ph15091039

**Published:** 2022-08-23

**Authors:** Cátia F. Marques, Maria Matilde Marques, Gonçalo C. Justino

**Affiliations:** 1Centro de Química Estrutural, Institute of Molecular Sciences, Instituto Superior Técnico, Universidade de Lisboa, 1049-001 Lisboa, Portugal; 2Departamento de Engenharia Química, Instituto Superior Técnico, Universidade de Lisboa, 1049-001 Lisboa, Portugal

**Keywords:** montelukast, leukotrienes, adverse drug reactions, repurposing

## Abstract

Increasing environmental distress is associated with a growing asthma incidence; no treatments are available but montelukast (MTK)—an antagonist of the cysteinyl leukotrienes receptor 1—is widely used in the management of symptoms among adults and children. Recently, new molecular targets have been identified and MTK has been proposed for repurposing in other therapeutic applications, with several ongoing clinical trials. The proposed applications include neuroinflammation control, which could be explored in some neurodegenerative disorders, such as Alzheimer’s and Parkinson’s diseases (AD and PD). However, this drug has been associated with an increasing number of reported neuropsychiatric adverse drug reactions (ADRs). Besides, and despite being on the market since 1998, MTK metabolism is still poorly understood and the mechanisms underlying neuropsychiatric ADRs remain unknown. We review the role of MTK as a modulator of leukotriene pathways and systematize the current knowledge about MTK metabolism. Known toxic effects of MTK are discussed, and repurposing applications are presented comprehensively, with a focus on AD and PD.

## 1. Introduction

Montelukast (MTK) is an antagonist of the cysteinyl leukotrienes receptor 1 and is routinely used in the management of asthma symptoms among adults and children. Its systemic anti-inflammatory actions, which are particularly important in the brain tissues, are at the onset of various clinical studies focused on the repurposing of this drug for various other diseases, aimed particularly at Alzheimer’s and Parkinson’s diseases. However, this repurposing clashes with neuropsychiatric adverse drug reactions elicited by the drug. Starting with a brief overview of the biochemistry of leukotrienes, this work reviews the current knowledge on montelukast.

## 2. Cysteinyl Leukotrienes—Multifunctional Inflammation Mediators

### 2.1. Cysteinyl Leukotrienes and Their Receptors

Initially described as the slow-reacting substances of anaphylaxis, leukotrienes (LTs) are pro-inflammatory lipid mediators derived from arachidonic acid [1,2]. These mediators are synthesized mainly in cells from the innate immune system (e.g., polymorphonuclear leukocytes, macrophages, mast cells, and brain microglia) following activation by immune and non-immune stimuli such as infection, tissue injury, allergens, and exercise (Figure 1). Upon cell activation, the cytosolic calcium concentration increases, and the cytosolic phospholipase A_2_ (cPLA_2_) and 5-lipoxygenase (5-LOX) enzymes are activated and translocated to the nuclear envelope. There, cPLA_2_ cleaves glycerophospholipids, releasing arachidonic acid (AA), which is converted to the acyclic hydroperoxide 5(*S*)-hydroperoxyeicosatetraenoic acid (5-HpETE) by 5-LOX-mediated oxidation upon LOX activation by 5-LOX activating protein (FLAP); 5-HpETE, in turn, undergoes dehydration to the unstable conjugated triene epoxide leukotriene A_4_ (LTA_4_), the first metabolite in the leukotriene pathway. LTA_4_ is a short-lived intermediate that can undergo conjugate addition of water to form leukotriene B_4_ (LTB_4_) or conjugation with glutathione by LTC_4_ synthase to form leukotriene C_4_ (LTC_4_, an *S*-glutathionyl LT). LTB_4_ and LTC_4_ are transported to the extracellular space mainly by multidrug resistance proteins, namely through MRP4 (LTB_4_) and MRP1 (LTC_4_) [3,4], where cleavage of LTC_4_ to leukotriene D_4_ (LTD_4_) and subsequently to leukotriene E_4_ (LTE_4_) takes place. LTD_4_, an *S*-cysteinyl LT, is synthesised from LTC_4_ by a γ-glutamyl transpeptidase (GGT)-mediated cleavage, whereas LTE_4_ results from the cleavage of LTD_4_ by a membrane-bound dipeptidase [5,6,7,8,9,10,11,12,13,14,15,16,17,18].

Figure 1 summarizes the biosynthesis of leukotrienes as well as their interactions with the leukotriene receptors.

LTB_4_ is a pro-inflammatory LT that acts on human polymorphonuclear leukocytes (PMNLs) such as neutrophils, via G protein-coupled receptors B-LT_1_ or B-LT_2_, triggering chemotaxis and the subsequent activation of the inflammatory response. LTC_4_, LTD_4_, and LTE_4_ constitute a group of cysteinyl leukotrienes (CysLTs) that act through G protein-coupled cell surface receptors, of which the two classical receptors are the cysteinyl leukotriene receptors 1 (CysLTR_1_) and 2 (CysLTR_2_). LTC_4_ is an agonist of CysLTR_1_ whereas LTD_4_ binds CysLTR_1_ and CysLTR_2_. LTE_4_ is described as an agonist of CysLTR_3_ (also known as GPR99 receptor) and of the purinergic receptors GPR17 and P2Y_12_ [5,6,7,8,9,10,11,12,13,14,15,16,17,18].

Cysteinyl leukotriene receptors (CysLTRs) are involved in the pathophysiology of various respiratory allergic diseases, including bronchial asthma, exercise- and aspirin-induced asthma, and allergic rhinitis, as well as atopic dermatitis, allergic conjunctivitis, and anaphylaxis, exhibiting a large overlap with the B-LT receptors, but allowing a finely tuned immune response [11,12,13,20,21]. Receptor engagement by CysLTs promotes bronchoconstriction, vascular leakage, and neutrophil extravasation to inflammation sites [7]. CysLTR_1_ is expressed in most human tissues, particularly in the appendix, oesophagus, gall bladder, lung, lymph nodes, spleen, and urinary bladder. The affinity of leukotrienes to this receptor varies in the order LTD_4_ > LTC_4_ > LTE_4_. This receptor is sensitive to classical antagonists (Figure 2) such as montelukast (MTK, Singulair^®^), zafirlukast (Accolate^®^), pranlukast (Onon^®^, Azlaire^®^), pobilukast, and MK571, all members of the *Lukast* group (cysteinyl leukotriene receptor antagonists).

CysLTR_2_ is predominantly expressed in the spleen, heart, brain, and adrenal gland, and its affinity strength is LTC_4_ = LTD_4_ > LTE_4_. HAMI3379 (Figure 2) was identified as a potent and selective CysLTR_2_ receptor antagonist [22]. To our knowledge, only two dual inhibitors of both CysTR_1_ and CysLTR_2_ are reported—BAY-u9773 and gemilukast (Figure 2). However, BAY-u9773 is neither very potent nor selective for human CysLTs [11,12,20,21,23] and gemilukast did not show outcome differences when compared with MTK [24,25]. Figure 2 also shows the experimental IC_50_ values available for these compounds.

**Figure 2 pharmaceuticals-15-01039-f002:**
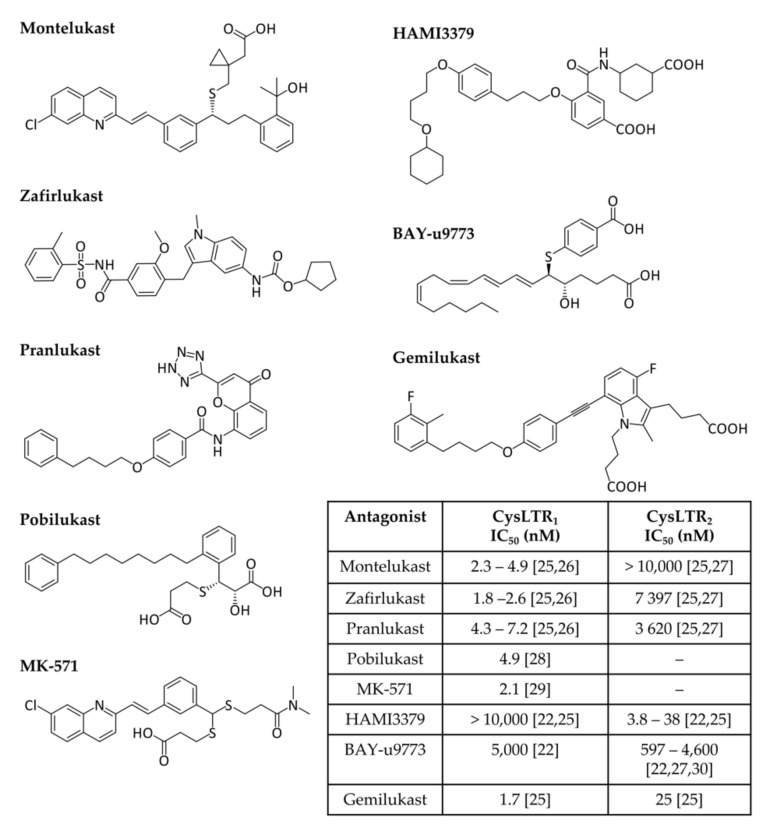
**Cysteinyl leukotrienes receptor antagonists.** CysLTR_1_ antagonists (montelukast, zafirlukast, pranlukast, pobilukast, and MK571), CysLTR_2_ antagonists (HAMI3379) and dual antagonists (BAY-u9773 and gemilukast) are shown. Experimental IC_50_ values available from the literature are also given in the inset table [22,25,26,27,28,29,30].

Besides these classical receptors, three other receptors are associated with the leukotriene cascade—GPR99, P2Y_12_, and GPR17.

GPR99, or OXGR1, is an α-ketoglutarate receptor that was originally thought to be a P2Y receptor [31]. This receptor is expressed in the kidney, placenta, trachea, salivary glands, lungs, and smooth muscle cells, as well as in some brain regions; in addition to its effects on acid–base homeostasis, it is also involved in axon growth [32,33,34,35]. GPR99 is considered the third CysLT receptor (CysLTR_3_) due to its high affinity for LTE_4_. No antagonists are currently available for this receptor [32,36].

P2Y_12_ is an adenosine diphosphate receptor that also mediates LTE_4_-dependent pulmonary inflammation (but not the LTD_4_ response) [37]. This receptor is mainly expressed in platelets and microglia, where it triggers platelet activation and blood clotting, and induces microglial chemotaxis in situations of central nervous system (CNS) injury [38,39,40,41]. P2Y_12_ is also associated with some asthma symptoms, namely with eosinophilic inflammation and airway hyper-responsiveness [42,43]. The P2Y_12_ receptor is blocked by anti-platelet drugs such as clopidogrel, prasugrel, and ticagrelor [44].

Lastly, GPR17 is a uracil nucleotide P2Y receptor expressed in the brain that also binds CysLTs [12,14,45,46,47,48,49,50]. This receptor is described as a sensor of neuronal damage, being activated by nucleotides and CysLTs released in the damaged area and plays a dual role depending on its surroundings: under physiological conditions, GPR17 contributes to the differentiation and maturation of oligodendrocytes, whereas under pathological conditions it mediates demyelination and apoptosis [51,52,53,54,55,56]. GPR17 is described as a putative negative regulator of CysLTR_1_ [57]. The CysLTR_1_ inhibitors pranlukast and montelukast are also antagonists of this receptor [46,58,59].

### 2.2. Leukotrienes in the Brain

The potential of leukotrienes as pro-inflammatory lipid mediators, described above, together with the pattern of expression of their receptors in different organs, has led to the suggestion that LTs play an important role in the central nervous system. In fact, recent advances have associated inflammation with some brain pathologies such as multiple sclerosis, Alzheimer’s disease, Parkinson’s disease, brain ischemia, and epilepsy, among others, and leukotrienes are thought to play a role in this process [60,61].

Despite having been originally found in leukocytes, leukotrienes are also present in the brain. Not only is the 5-LOX enzyme widely distributed in various brain regions (e.g., cortex, hippocampus, and cerebellum), but CysLTs are also produced by vascular endothelial cells, neurons, and glial cells upon LTA_4_ expression by activated neutrophils [47]. CysLTR_1_ is widely expressed in the cortex, hippocampus, and nigrostriatum, as well as in cerebrovascular endothelial cells, astrocytes, microglia, and several types of neurons. On the other hand, CysLTR_2_ is expressed in the cortex, hippocampus, substantia nigra, astrocytes, microglia, and neurons [62,63,64,65,66]. These receptors are usually weakly expressed unless activated by pathological stimuli [66]. Some studies have shown that the exposure of neurons to acute neuronal injury is associated with upregulated levels of CysLTR_1_ and CysLTR_2_, and with increased blood–brain barrier (BBB) permeability. Once activated, CysLT receptors will trigger an inflammatory cascade, activating pro-inflammatory cytokines and inflammation, ultimately leading to neuronal damage [62,64,65,67].

#### 2.2.1. Leukotrienes: Role in Neuroinflammation

Neuroinflammation is a complex biological response of the brain and spinal cord mediated by the production of pro-inflammatory cytokines (IL-1β, IL-6, and TNF-α), chemokines (CCL2, CCL5, and CXCL1), reactive oxygen species (ROS), and other mediators (NO, prostaglandins, and leukotrienes) [68,69,70]. This biological response is associated with restoration of homeostatic balance, in order to eliminate and repair the initial cause of cell injury, and can be classified as acute (seconds to days) or chronic [68,71].

An acute inflammatory response is an adaptive response, usually beneficial, meant to protect tissues from a specific injury as trauma or infection [71]. As presented in Figure 3, in situations of acute inflammation, the immune system priorities are neuroprotection, tissue repair, and neuroplasticity. When the brain is exposed to immune signals after any infection, microglia and astrocytes are activated and neuroinflammatory cytokines such as IL-1β, TNF-α, and IL-6 are expressed to sustain the inflammatory response. This response is short and transient, and no severe effects take place [68]. Brain development and plasticity are other positive aspects of neuroinflammation. Neurons, astrocytes, and glia cells are involved in neurotransmission through the modulatory effect of cytokines and neuromodulators such as IL-1β, IL-6, TNF-α, NF-κB, and glutamate [68,72]. Brain tissue repair can also be stimulated through the activation of macrophages, lymphocytes, and microglia, which promotes angiogenesis, axon regeneration, myelin clearance, and oligodendrocyte regeneration [68,73,74,75]. Lastly, immune system training through immune pre-conditioning or euflammation allows modulation of the microglia response against hyper-inflammatory conditions, protecting the brain from CNS injuries [68,76,77].

However, if the acute inflammation response fails and the inflammation process persists, chronic inflammation ensues with a long-lasting maladaptive or defective response that could destroy tissues and compromise the immune response (Figure 3) [70,71]. Characteristically, there is an increased production of cytokines (IL-1 and TNF-α), reactive oxygen species (ROS), and other inflammatory mediators (e.g., inducible nitric oxide synthase, iNOS), associated with the activation of microglia cells, and consequent expression of more pro-inflammatory cytokines and chemokines in the brain [68]. This activation could be caused by noradrenergic signalling, inflammasome activation, and ATP release [78,79,80]. Microglia activation is also involved in the recruitment of monocytes from the bone marrow to the brain and is linked to anxiety-like behaviour and to the development of mood disorders [68,81].

The normal ageing process is one example of the disruption of the communication pathways between the brain and the immune system, leading to chronic neuroinflammation. During ageing, there is an increase in inflammatory (e.g., IL-1β and IL-6) and a decrease in anti-inflammatory (e.g., IL-10 and IL-4) cytokines that results in damage to the nervous system and the onset of neurodegenerative diseases [68].

It has been shown that the leukotriene receptors CysLTR_1_ and CysLTR_2_ in different brain cells, namely microglia (known as the brain’s immune system), astrocytes, and several types of neurons, are upregulated in response to brain injury such as brain ischemia, Alzheimer’s disease, and Parkinson’s disease [62,63,64,65,82,83,84,85,86,87,88]. The modulation of these receptors is associated not only with the outcome of acute inflammation but also with the restoring of homeostasis during chronic inflammation [62,63,64,65,82,83,84,85,86,87,88].

Although the mechanisms of action are still poorly understood, evidence supports the relationship between leukotrienes and neuroinflammation, suggesting the use of leukotriene antagonists as a possible therapeutic strategy in neuroinflammation, given that antagonists of either CysLTR_1_ or CysLTR_2_ display wide multi-target anti-inflammatory activity [66]. Both receptors are expressed at low levels in multiple brain regions, but are upregulated following injury, as observed in various experimental models of ischemia and Alzheimer’s and Parkinson’s diseases [65,82,83,84,89]. Interestingly, silencing the expression of the genes coding for these two receptors leads to in vivo protection against lipopolysaccharide- and ischemia-induced brain inflammation and injury [87,88]. Although this strategy needs to be further explored, it could be a very promising therapeutic approach to the improvement of symptoms (or even disease treatment) in patients who suffer from neurodegenerative disorders and have no alternative therapy to manage the debilitating symptoms characteristic of neurodegeneration.

#### 2.2.2. Leukotrienes in Neuro-Signalling Pathways

Message transmission between neurons results from an electrical impulse (action potential) that causes the release of neurotransmitters into the synaptic cleft. After crossing the synaptic cleft, neurotransmitters will reach their receptors on the postsynaptic side to excite or inhibit the target neuron. Excitatory synaptic transmission is mainly assured by L-glutamate, whereas γ-aminobutyric acid (GABA) is the major neurotransmitter involved in the inhibitory synaptic response. In addition to these neurotransmitters, there are other molecules involved in signalling and neuromodulation, such as acetylcholine, monoamines (e.g., dopamine, adrenaline, serotonin, and histamine), purines (e.g., adenosine), and neuropeptides [89].

A close relationship between neuroinflammation and neuro-signalling pathways has been proposed. One example is the involvement of excitotoxicity in neuroinflammation: an exacerbated or prolonged activation of glutamate receptors, particularly the *N*-methyl-D-aspartic acid receptors (NMDA), causes an increase in calcium influx into the neurons. This increase of intracellular calcium levels leads to a neurotoxic response, including the activation of the AA pathway, that can lead to the loss of neuronal function and, ultimately, cell death [90]. Studies involving CysLTR antagonists showed that pranlukast was able to inhibit NMDA-induced CysLTR_1_ expression, leading to a decrease in excitotoxic cell death [91]. Montelukast also presented a strong anti-excitotoxicity effect, as well as anti-inflammatory and neuroprotective properties [83].

Dopamine reuptake is also associated with the leukotriene pathway. Inhibition of the 5-LOX activating protein (FLAP) is associated with the improved integrity of dopaminergic neurons [92].

#### 2.2.3. The Leukotriene Link between Stress and Depression

As suggested in Figure 3, depression can result from chronic neuroinflammation. Not only pro-inflammatory cytokines (e.g., IL-1β and TNF-α) were found to be dysregulated in depression patients, but also IL-1β, IL-6, TNF-α, or lipopolysaccharide (LPS) administration in animal models led to depression- and anxiety-like behaviours [93,94,95,96].

Stress stimuli led to an increase in calcium concentration, releasing AA after cPLA_2_ activation by phosphorylation [97]. Once released, AA is used to synthesise leukotrienes (Figure 1) and prostaglandins. A study using mice in which the *cysltr1* gene was silenced in the hippocampus suggested that the absence of CysLTR_1_ prevents the development of neuroinflammation and of a depressive-like phenotype [98]. The effects observed upon blocking the same receptors in a mouse lipopolysaccharide-induced neuroinflammation model support those previous results [99]. Inhibition of the 5-LOX enzyme has also been associated with a relief of depression-like behaviour [100].

#### 2.2.4. The Role of Leukotrienes in Neurodegenerative Diseases

Besides their role in inflammation, leukotrienes are also involved in some of the most characteristic hallmarks of neurodegenerative disorders (Figure 4): neuronal cell death, neuroinflammation, altered neurogenesis, and disrupted blood–brain barrier and vascular system, among others.

The clear association between neuroinflammation and Alzheimer’s and/or Parkinson’s disease led to the study of the role of CysLTs pathways and receptors in these diseases.

Alzheimer’s disease (AD, described in more detail in Section 4.1) is a neurodegenerative disease characterized by memory loss and dementia. There is evidence for CysLTR_1_ involvement in AD, leading to amyloidogenesis and neuroinflammation. In particular:

(1) In an AD mouse model (APP/PS1 double transgenic, overexpressing mutated forms of human amyloid precursor protein, APP, and presenilin 1), the expression of CysLTR_1_ was found to increase with ageing, and to correlate with Aβ deposits and behaviour deficits [84,101];

(2) LTD_4_ upregulates APP, β-, and γ-secretase levels, and facilitates Aβ amyloid accumulation via the CysLTR_1_-mediated NF-κB pathway [102,103,104].

Aggregated Aβ_1–42_ is known to cause AD-like neurotoxicity and cognitive deficiency, associated with pro-inflammatory cytokine production (TNF-α, IL-1β) and increased cell apoptosis [84,105,106]. Additional studies also revealed that Aβ plaques are associated with an increased oxidative stress status. Oxidative stress is known to upregulate cPLA_2_ activity, leading to an increased release of arachidonic acid metabolites [66]. These responses are inhibited by *Lukast* drugs (pranlukast, montelukast, and zafirlukast), suggesting that CysLTR_1_ is a pro-inflammatory regulator and is involved in AD initiation and progression [66,84,105,106].

Parkinson’s disease (PD, described in more detail in Section 4.2) is also a neurodegenerative disorder characterised by the progressive degeneration and loss of dopaminergic neurons. Inflammation induction in PD models (with rotenone or lipopolysaccharide) leads to microglia activation, increasing the production of the pro-inflammatory cytokines TNF-α, IL-1β, and IL-6, and brain inflammation, leading to dopaminergic neuronal loss [47,66,107,108,109,110]. This action was inhibited by montelukast via the CysLTR_1_-mediated p38 MAPK/NF-κB pathway [82,107,111], and also by selective inhibition or knockout of CysLTR_2_ [86], suggesting that CysLTR_1_ and CysLTR_2_ could be strategic targets against PD. CysLTR_1_, as well as 5-LOX, are found to be upregulated in mouse PD models [92], further strengthening the hypothesis that the LT pathway contributes to the progression of PD.

In conclusion, leukotrienes play an important role in the progression of neurodegenerative disorders. Receptors involved in the different steps of the LT cascade interfere with the inflammatory process, which is partially responsible for the development of the characteristic hallmarks of AD and PD. For this reason, targeting the CysLT pathway seems to be a promising strategy to delay the progression of these disorders.

## 3. A Cysteine Leukotriene Receptor Antagonist Known as Montelukast

The World Health Organization (WHO, Geneva, Switzerland) estimated that in 2019 more than 262 million people suffered from asthma, a pulmonary disorder that causes lung inflammation and tightening of the muscles around small airways [112]. The number of people suffering from this disorder is expected to increase, since a wide range of environmental risks is associated with asthma development, including tobacco smoking, pollution, and environmental allergens and irritants [112]. With no current treatment available, montelukast is broadly used in symptom management in adults and children.

Montelukast (MTK, 1-([(1(*R*)-(3-(2-(7-chloro-2-quinolinyl)-(*E*)-ethenyl)phenyl)-3-(2-(1-hydroxy-1-methylethyl)phenyl)propyl)thio]methyl) cyclopropylacetic acid, Figure 5), widely used in asthma management and allergic rhinitis, is a potent antagonist of CysLTR_1_, a receptor with high affinity for the leukotriene, LTD_4_. As indicated above, CysLTRs modulate the synthesis of the leukotrienes from arachidonic acid. They are involved in the pathology of various allergic diseases of the respiratory system, including bronchial asthma, exercise- and aspirin-induced asthma, and allergic rhinitis. These lead to airway constriction, smooth muscle contraction, and alterations in the inflammatory processes, such as neutrophil extravasation to the site of inflammation [113,114,115,116].

In addition to CysLTR_1_, targeted in asthma management, recent studies have identified further MTK targets that could be exploited against other pathologies, particularly in the central nervous system. MTK has been identified as an inhibitor of 5-LOX [117], and as an antagonist of the CysLTR_2_, P2Y_12_ [118], and GPR17 [47] receptors.

### 3.1. Montelukast Metabolism and Bioavailability

The sodium salt of MTK has been available since 1995 as Singulair^®^ (Merck Sharp & Dohme, Kenilworth, NJ, USA) and has been increasingly prescribed in recent years [119]. However, MTK metabolism is still poorly understood.

The first metabolic studies of this drug were performed on healthy volunteers treated with ^14^C-MTK. Samples from blood, urine, faeces, bile, and gastric juices were collected during the clinical trial and eight MTK metabolites were identified (Figure 5) an acyl glucuronide (M1), a sulfoxide (M2), a phenol (M3), a dicarboxylic acid (M4), and hydroxylated metabolites at positions 21 (M5a/b, where a and b correspond to the 21-*S* and -*R* configurations) and 36 (M6a/b, with no assignment of the specific C36 configuration) [120]. According to the authors of the study, MTK is mainly excreted in the faeces (86% of the administered dose) and only 0.2% is excreted via urine. All identified metabolites were found in bile samples, where M5a was more abundant than M5b. M5 and M6 were also identified in plasma samples, with M6a being more abundant than M6b [120].

Clinical trial data were complemented with early in vitro studies, which showed that cytochrome P450 (CYP) enzymes are responsible for the phase I MTK metabolism, whereas flavin-containing monooxygenases (FMO) have little or no activity on this substrate [121,122].

Additional metabolic and inhibition assays have been described to identify further metabolites formed during MTK metabolism (phase I and II), as well as the active isoforms. Human liver microsomes, recombinant CYP enzymes (CYP1A2, 2A6, 2B6, 2C8, 2C9, 2C19, 2D6, 2E1, 3A4, and 3A5), UGT enzymes (UGT1A1, 1A3, 1A4, 1A6, 1A7, 1A8, 1A9, 1A10, 2B4, 2B7, 2B10, 2B15, and 2B17), and isoform-specific inhibitors were tested [120,121,122,123,124]. Only a new MTK ether glucuronide metabolite (M-glucuronide, Figure 5) was found in those studies.

The major enzymes responsible for metabolite production are identified in Figure 5. Table 1 shows the relative contribution of each CYP isoform toward the final products. CYP2C8 is mentioned to be the most relevant CYP involved in MTK metabolism, responsible for 70% of MTK oxidative metabolic clearance [123].

**Figure 5 pharmaceuticals-15-01039-f005:**
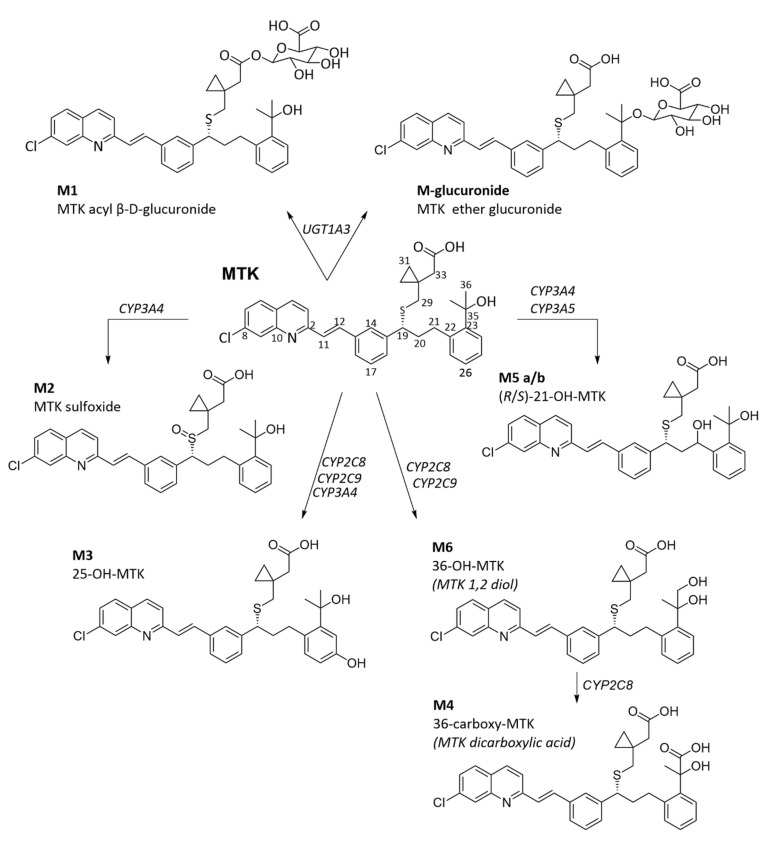
Human MTK metabolic pathways. The presented metabolites were identified in bile (M1, M-glucuronide, M2, M3, M4, and M6) and in plasma (M5 and M6) from healthy volunteers [120,121,122,123,125]. Atom numbering used in this work is included, in agreement with the literature numbering system used for montelukast.

Briefly, MTK sulfoxide (M2) is produced mainly by CYP3A4 whereas M3 (25-OH-MTK) is obtained by the action of the CYP2C8, CYP2C9, CYP3A4, and CYP2C19 isoforms [121,122,123,124]. M5 (21-OH-MTK) results from CYP3A4 and CYP3A5 metabolism [121,122,123,124]. The dihydroxylated metabolite (M6, 36-OH-MTK), precursor to the dicarboxylate (M4), is produced by CYP2C8 and CYP2C9 [121,122,123,124], whereas M4 itself results from CYP2C8 catalysis [123]. M4 was considered to be the major metabolite by Balani et al. [120], whereas Cardoso et al. [124] consider M6 as the most abundant. For VandenBrink et al. [122], M2, M5, and M6 are the major ones. However, M2 is not a consensual metabolite since it is also an MTK impurity [123,124].

Only two phase II MTK metabolites have so far been identified by mass spectrometry: the M1 glucuronide (major) and the M-glucuronide (minor), both stemming from glucuronidation of the parent drug. UGT1A3 is reported to be the most active UDP-glucuronosyltransferase isoform involved in MTK glucuronidation [124,125].

MTK is characterized by a 60–70% bioavailability and high plasma protein binding capacity (>99%) and displays little or no gender effect on its pharmacokinetic properties [126,127]. No metabolism differences are reported between adults and children [121].

Despite the unfavourable absorption properties, such as high lipophilicity and high plasma protein binding, the efficiency of MTK transport across membranes, including the blood–brain barrier, remains unknown. Even though some researchers have identified MTK as a substrate of the organic anion transporting polypeptide 2B1 (OATP2B1) transporter, which is expressed in the blood–brain barrier, recent studies failed to confirm this observation [128,129,130,131].

### 3.2. Adverse Drug Reactions Related to Montelukast Administration

Montelukast belongs to the *Lukast* drug family, whose members are considered safe and well-tolerated drugs, suitable for long-term administration, with low toxicity and relatively low adverse side effects [116]. However, during post-marketing surveillance, some reports of adverse effects caused by monotherapy and co-adjuvant therapy with MTK emerged, motivating the US FDA to require a boxed warning regarding montelukast use and the occurrence of neuropsychiatric events. Since March 2020, the FDA recommends the use of alternative drugs, restricting MTK to patients with an inadequate response or intolerance to other therapies [132,133,134].

According to the WHO global database for adverse drug reactions (ADRs), Vigibase [135], 26,253 reports were filed until August 2021, and 22% of the reported ADRs occurred in children between the ages of 2 and 11. Psychiatric and nervous system disorders are the most reported, along with hepatobiliary, pancreatic, and uropoietic disorders, and immune system dysregulation. Although the number of reported ADRs is considerable, no underlying mechanisms have been proposed.

It is important to highlight the limitations of the ADR reporting system. Since ADR reporting is voluntary and patients and doctors only report when a correlation between a drug and ADRs is suspected, it is expected that MTK-related ADRs are underreported. On the other hand, some of the neuropsychiatric events experienced by patients are not exclusively correlated with MTK, but also with other physiopathologic, economic, and social conditions such as, for example, depression and sleep deprivation.

MTK toxic events are described in more detail in the following sections.

#### 3.2.1. Neuropsychiatric and Nervous System Disorders

A growing number of MTK ADRs has been reported in the literature, focusing on neuropsychiatric aspects, especially anxiety and sleep disorders [136,137,138,139,140,141,142].

In 2009, a total of 48 reports of psychiatric disorders in children were found in the Swedish ADR database SWEDIS. Nightmares, general anxiety, aggressiveness, sleep disorders, insomnia, irritability, hallucination, hyperactivity, and personality disorder were some of the most reported ADRs. Approximately 50% of these effects occurred in children under 3 years old and, in 80% of the reports, ADRs developed within 1 week from the first MTK administration [143]. Later, a cohort of 14,670 individual case safety reports, of which 2630 corresponded to children and adolescents younger than 18 years old, were reviewed in 2015. The main conclusions highlighted children as the most likely to experience montelukast ADRs: sleep disorders were mostly reported in children younger than 2 years old; depression and anxiety signs in children between 2 and 11 years; and suicidal behaviour and depression/anxiety in adolescents between 12 and 17 years. Surprisingly, achieved suicides were more reported in children than adolescents or adults [137]. Between 2012 and 2017, an observational study in a Spanish paediatric hospital concluded that 5.7% of children under 15 years old experienced ADRs, mainly insomnia, hyperactivity, and nightmares, which disappeared after MTK discontinuation [144].

Isolated cases of well-defined neuropsychiatric events in children and adults taking montelukast are also described in the literature. A 9-year-old boy experienced sleepwalking, sleep disturbance, bruxism, and anxiety during MTK treatment. After MTK withdrawal, the symptoms resolved without further intervention [139]. Another case described a 13-year-old who experienced hallucinations that stopped 48 h after MTK withdrawal [138]. A 16-year-old girl who was medicated with MTK reported parasomnias (sleeptalking and sleepwalking) during two attempts at MTK treatment. Symptoms stopped after MTK withdrawal for both attempts [145]. A 29-year-old asthmatic woman suffered from visual and auditory hallucinations, which stopped two days after MTK withdrawal [146]. An HIV-positive female patient reported neuropsychiatric disturbance including sleep disorders, vivid dreams, irritability, and confusion after adding MTK to her usual medication (efavirenz) [147]. In this case, doctors suspected drug–drug interaction and drug competition between CYP isoforms involved in MTK metabolism [147].

The association between MTK and ADRs is in permanent evaluation. A recent study reports that the risk for psychiatric adverse events is greater in patients with past psychiatric history, and no association between depression or self-harm events and hospitalizations was identified [134]. Regarding MTK and suicide ideation, it remains a non-consensual subject among the scientific community, with some studies establishing a relationship between them and others denying it [148,149,150]. There is even a study suggesting that MTK may reduce the risk of suicide [151].

Summing up, both children and adults seem to develop psychiatric adverse side effects during montelukast treatment, especially children. Usually, the symptoms tend to disappear after drug withdrawal. However, it is important to understand the mechanisms underlying these ADRs in order to improve treatment and risk–benefit assessment, and to prevent dangerous outcomes.

#### 3.2.2. Hepatobiliary, Pancreatic, and Uropoietic Disorders

Contrary to the neuropsychiatric ADRs, the hepatotoxicity of montelukast occurs mainly in adults. Usually, patients are polymedicated and the relationship between MTK and ADRs was established based on time exposure and drug exposure and withdrawal.

Patients between 22 and 76 years old medicated with MTK developed acute pancreatitis, hypercholesterolemia, and hypertriglyceridemia [152], haematuria [153], and hepatomegaly [154]. Vomiting, icterus, and high levels of liver biomarkers (aminotransferase, bilirubin, and alkaline phosphatase) were also associated with MTK treatment, with an underlying related immune-mediated mechanism of liver injury [155]. All patients improved their condition after MTK withdrawal.

Regarding children, there are clinical cases describing hepatitis, nausea, vomiting, abdominal pain, and high levels of liver biomarkers [156], as well as hepatocellular injury [157]. Children also recovered after MTK withdrawal.

With these examples in mind, kidney and renal function should be monitored in patients on montelukast therapy. The risk of hypertriglyceridemia can be harmful not only for cardiovascular risk patients but also for healthy patients [133].

#### 3.2.3. Skin and Subcutaneous Tissue Disorders

Angioedema and urticaria, conditions with a strong inflammatory component, are the most commonly reported skin disorders in patients medicated with MTK [158,159]. A case of a child with erythematous and bullous eruption in the lower extremities after MTK treatment was also reported [160]. In all cases, symptoms promptly resolved upon stopping the treatment with MTK; in one case, symptom reappearance upon MTK re-introduction clearly established a link between MTK and the observed side effects [159].

#### 3.2.4. Immune System Disorders

Immune system disorders associated with montelukast therapy are rare and include anaphylaxis (very rare), hepatic eosinophilic infiltration, and autoimmune vasculitis [133].

Churg–Strauss Syndrome (CSS), also known as allergic granulomatous angiitis, has been reported in adult patients, and no cases seem to have been reported in children [161,162]. MTK treatment has been associated with a 7.5-fold higher risk of developing CSS [163,164,165,166]. This syndrome is a rare vasculitis disorder of small- and medium-size vessels and could be characterized by blood eosinophilia and eosinophilic infiltration into affected tissues [167]. Patients who experienced CSS developed eosinophilia, leucocytosis, pulmonary infiltrates, malaise, fever, rash, neuropathy, and biomarker alterations (e.g., antineutrophil cytoplasmatic antibody and serum bilirubin) [168,169,170,171,172,173,174,175,176].

Henoch–Schlönlein syndrome affects mainly male children between 3 and 15 years old and is characterized by a tetrad of clinical manifestations including palpable purpura, arthritis–arthralgia, abdominal pain, and renal disease [177]. All secondary effects disappeared on MTK removal.

#### 3.2.5. Montelukast Administration during Pregnancy

Maternal asthma has been associated with an increased risk of pregnancy complications, including pre-eclampsia, vaginal haemorrhage, pregnancy-induced hypertension, and low birth weight [178]. Currently, montelukast is classified as a category B drug in pregnancy risk information (no evidence of risk is associated), with limited information available.

Despite the identification of limb reduction defects in live-born offspring from mothers treated with MTK [179] during the post-marketing surveillance phase by Merck (1997–2006), there are not enough studies to support or refute the possible causes of these observations. In fact, several studies claimed no association between MTK and teratogenicity or risks of adverse prenatal outcomes [179,180,181]. Since preterm birth and maternal complications (preeclampsia and gestational diabetes) are also associated with asthma [182], conclusions regarding MTK safety during pregnancy should be interpreted carefully.

To summarize, neuropsychiatric adverse side effects are the most reported ones and occur mainly in children. Hepatotoxicity should also be monitored with care, as well as the possible occurrence of immune responses after montelukast exposure.

### 3.3. Montelukast Repurposing Applications

Recently, montelukast has been proposed for repurposing in other therapeutic applications (Table 2), with several of these potential uses already undergoing clinical trials. A mid-2021 search of the NIH Clinical Trials Database (clinicaltrials.gov) identified 29 clinical trials using MTK in the treatment of various pathologies such as bronchiolitis, osteoarthritis, rheumatoid arthritis, pain, Alzheimer’s disease, obesity and diabetes, steatohepatitis, and dengue.

Concerning the central nervous system, MTK has been suggested as a potential drug against some neurogenerative disorders, including Alzheimer’s disease [47,105,183], Parkinson’s disease [82,107,111,184,185], and Huntington’s disease [186]. Montelukast seems to be able to improve cognitive and neurological functions due to its modulation role in the inflammatory and apoptotic cascades involved in neurodegenerative features, particularly those where TNF-α, NF-κB, caspase-3, Bcl-2, MAPK, and IL-1β participate. These are among the most relevant signalling proteins involved in neurodegeneration.

Additionally, MTK also appears to lead to a decrease in α-synuclein load and in Aβ_1__–__42_-induced neurotoxicity. MTK has also been found to modulate the oxidative stress associated with a dysregulation of the GSH/GSSG balance or of superoxide dismutase activity, two key factors in the maintenance of redox homeostasis [47,82,105,107,111,183,186,187,188,189,190].

The application of MTK as a chemopreventive and adjuvant agent in cancer therapy has been suggested by different research teams [191,192,193,194,195]. Previous results show that MTK is able to induce cancer cell death by inhibiting cell proliferation, downregulating Bcl-2, and promoting nuclear translocation of the apoptosis-inducing factor (AIF) [191]. The downregulation of the hypoxia-inducible factor-1α (HIF-1α) [193] has also been mentioned as a mechanism targeting cancer cells, as well as the inhibition of the TNF-α-dependent IL-8 expression and the suppression of the NF-kB p65-associated histone acetyltransferase activity (HAT) activity [195].

During the COVID-19 pandemic, MTK was used as an off-label drug in the prevention and treatment of pulmonary distress in patients infected with SARS-CoV-2. Its properties as an anti-inflammatory drug allied to cardiovascular benefits on thrombosis and vascular damage, as well as the potential beneficial effects on brain functions, make this drug a good candidate against COVID-19 symptoms [196,197,198,199,200].

Table 2 summarizes the repurposing applications that have been published regarding the use of montelukast in pathologies other than asthma and allergic rhinitis. Although new applications include conditions such as cancer, cardiovascular diseases, and neurodegenerative disorders, proposed applications involving CNS pathologies represent more than 50% of the available data.

**Table 2 pharmaceuticals-15-01039-t002:** Repurposing applications proposed for montelukast. Montelukast has been proposed for repurposing in different therapeutic applications, including cancer, degenerative disorders, and renal failure.

Models	Modulation	Outcome	
Bones and joints			
**C57B/6 mice with a femoral fracture**	Pharmacological treatment with MTK	↑ chondrocyte proliferation and early bone formation	[201]
**In vitro osteoarthritis model with chondrocytes (ATDC5)**	Pharmacological treatment with MTK	↓ cartilage degradation; ↓ cell injury, oxidative stress, apoptosis; ↓ CysLTR_1_ expression; ↑ KLF2 expression	[202]
**Cancer**			
**Nationwide population-based study with data from the Taiwan National Health Insurance Research Database**	Cancer patients with diagnosed asthma, treated with leukotriene inhibitors	↓ cancer risk	[192]
**Human lung cancer cells and Lewis lung-carcinoma-bearing mice**	Pharmacological treatment with MTK	Cell proliferation inhibition; ↓ Bcl-2; ↑ Bak; ↑ nuclear translocation of AIF; ↓ phosphorylation of WNK1, Akt, Erk1/2, MEK, and PRAS40 proteins	[191]
**Prostate cancer cell lines**	Pharmacological treatment with MTK	↓ HIF-1α protein; ↑ phosphorylation of eIF-2α	[193]
**Phorbol-myristate--acetate-differentiated U937 cells**	Pharmacological effect of MTK	↓ TNF-α-stimulated IL-8 expression; no effect on NF-kB p65 activation; suppressed NF-kB p65-associated HAT activity	[195]
**Tumour specimens from patients with prostate cancer and prostate cancer cell lines**	Pharmacological treatment with MTK	CysLTR_1_ overexpressed in prostate tissues; ↑ apoptosis of prostate cancer cells	[194]
**Cardiovascular**			
**Nationwide population-based study (Swedish population)**	Association between MTK use and cardiovascular outcomes	↓ recurrent cardiovascular events	[203]
**Nationwide population-based study (Swedish population)**	Association between MTK use and cardiovascular outcomes	↓ risk of aortic stenosis	[204]
**Asthmatic patients**	Pharmacological effect of MTK on cardiovascular risk	↓ levels of cardiovascular disease-associated inflammatory biomarkers and lipid levels	[205]
**CNS: Alzheimer’s disease**			
**Transgenic** **5xFAD Mice (AD mouse model)**	Pharmacological effect of MTK on neuroinflammation (microglia and CD8^+^ T cells)	↑ Tmem119+; ↓ genes related to AD-associated microglia; ↓ infiltration of CD8^+^ T-cells into the brain parenchyma; ↑ cognitive functions; ↓ 1061 genes (e.g., *Gpr17, Entpd1, Mlec*); ↑ 744 genes (e.g., *Zfp46, Ciart, Dbp*); more pronounced effect in females	[189]
**Transgenic *DCX-DsRed2* and wildtype Fisher 344 rats, FoxO1/3/4^fl^ mice** **, and** **GPR17** ** ^_^ ** **/** ** ^_^ ** ** ^GFP^ ** **mice**	Pharmacological treatment with MTK	↑ learning and memory in old rats; no effect on learning in young rats; ↓ microglia inflammation; ↑ BBB integrity; ↑ hippocampal neurogenesis; ↓ GPR17; ↓ CD68; ↑ claudin-5; ↑ PCNA, DCX, NeuN	[47]
**Intracerebroventricular infusions of aggregated Aβ_1–42_ in ICR mice**	Rescue effect of MTK on Aβ_1–42_-induced neurotoxicity	↓ memory impairment; ↓ inflammation and apoptosis markers; ↓ CysLTR_1_ mRNA/protein; ↓ IL-1β, TNF-α, NF-κB p65; ↓ caspase-3; ↑ Bcl-2	[105]
**Primary mouse neurons (foetal ICR mice) treated with Aβ_1–42_**	Rescue effect of MTK on Aβ_1–42_-induced neurotoxicity	↑ cell viability; ↓ CysLTR_1_ mRNA/protein; ↓ IL-1β, TNF-α; NF-κB p65; ↓ caspase-3; ↑ Bcl-2	[183]
**Intracerebroventricular streptozotocin-induced model of sporadic AD in ICR mice**	Pharmacological treatment with MTK	↓ memory impairment; ↓ neuroinflammation and apoptosis; ↓ CysLTR_1_ expression; ↓ TNF-α, IL-1β, NF-κB p65; ↓ cleaved caspase-3; ↑ Bcl-2/Bax ratio	[190]
**CNS: Anti-nociception**			
**Local antinociception model of pain**	Pharmacological treatment with MTK	↓ local pain behaviour in both phases (neurogenic and inflammatory); Involvement of L-Arg/NO/cGMP/K_ATP_ channel pathway and PPARγ receptors	[206]
**CNS: Brain ischemia**			
**Middle cerebral artery occlusion model in mice and rats**	Pharmacological treatment with MTK	↓ behavioural dysfunction, brain infarct volume, brain atrophy, and neuron loss	[207]
**Bilateral carotid artery occlusion model in rats**	Pharmacological prophylaxis and treatment with MTK	↓ oxidative stress, inflammatory and apoptotic markers (myeloperoxidase, NF-κB, TNF-α, and IL-6); ↓ glutamate and lactate dehydrogenase	[83]
**CNS: Dementia with Lewy bodies**			
**Human brain specimen and female transgenic mice expressing human wild-type α-synuclein vs. their wild-type litter mates**	Pharmacological treatment with MTK	↑ memory function; ↓ α-synuclein load in the dentate gyrus; ↑ Beclin-1 expression; autophagy as a possible mechanistic pathway	[187]
**CNS: Epilepsy**			
**Epilepsy-induced spontaneous recurrent seizures with pentylenetetrazole (PTZ) in mice**	Pharmacological treatment with MTK	Prevention of PTZ-induced BBB disruption; ↓ recurrent seizures; ↓ mean amplitude of electroencephalography recording during seizures	[208]
		↓ recurrent seizures; ↓ frequency of daily seizures	[209]
**Pilocarpine-induced seizures in mice**			
**Electrically-induced seizures in mice**			
**CNS: Huntington’s disease**			
**Intrastriatal-quinolinic-acid-and malonic-acid-induced Huntington’s-like symptoms in rats**	Pharmacological treatment with MTK	↓ behavioural alterations; ↓ oxidative stress; ↓ mitochondrial dysfunction; ↓ TNF-α level	[186]
**CNS: Multiple Sclerosis**			
**MOG_35-55_-induced experimental autoimmune encephalomyelitis in female mice**	Pharmacological treatment with MTK	↓ CNS infiltration of inflammatory cells; ↓ clinical symptoms; ↓ IL-17; ↓ BBB disruption	[210]
**CNS: neurological ageing**			
**Observational study using data from two databases: NorPD and the Tromsø Study**	Association between MTK use and neurological health	Improved cognitive and neurologic function	[188]
**CNS: Parkinson’s disease**			
**Rotenone-induced model of PD in rats**	Pharmacological treatment with MTK	↑ locomotor activity; ↓ immobility time; ↓brain MDA levels; ↑ GSH levels; ↓ TNF-α levels	[111]
		↑ locomotor activity; ↓ p38 MAPK, TNF-α, IL-1β, NF-κB; ↓ CysLTR_1_ expression; ↓ p53 mRNA, caspase-3; ↑ GSH, SOD; ↓ MDA levels	[82]
**6-Hydroxydopamine mouse model (C57BL/6 mice) of PD**	Therapeutic effects of MTK	↓ TNF-α levels; ↓ IL-1β	[107]
**COVID-19**			
**Computational methods**	Target-based virtual ligand screening and molecular docking	Well-fitted in the active pocket of SARS-CoV-2 3CLpro, Mpro and RdRp	[211,212]
**Retrospective study of COVID patients**	COVID patients treated with or without MTK	↓ events of clinical deterioration	[213]
**Glaucoma**			
**Magnetic microbead injection into the anterior chamber of female Brown Norway rats**	Pharmacological treatment with MTK	↓ intra ocular pressure; ↑ retinal ganglion cell survival in ocular hypertension eyes; ↓ activation of Iba1^+^ microglial cells in retina; ↓ GPR17^+^ cells	[214]
**Lung transplant**			
**Bronchiolitis obliterans syndrome after lung transplantation in patients**	Pharmacological treatment with MTK	↓ forced expiratory volume in 1 s (FEV_1_)	[215,216,217]
**Pulmonary fibrosis**			
**Bleomycin-induced pulmonary fibrosis in female C57BL/6J mice**	Pharmacological prophylaxis and treatment with MTK	↓ fibrotic area; ↓ IL-6, IL-10, IL-13, and TGF-β1 mRNA levels; ↑ CysLTR_2_ mRNA expression	[218]
**Renal failure**			
**Rhabdomyolysis-induced acute renal failure in Wistar rats**	Pharmacological prophylaxis and treatment with MTK	Improved functional and structural renal damage; ↓ tubular damage; ↓ serum creatinine and urea levels; ↓ serum phosphate levels; ↓ GSH and MDA levels; ↑ SOD levels; ↓ serum TNF-α, TGF-β1, Fas, IL-10; ↑ IL-6/ TNF-α ratio	[219]
**Cisplatin-induced renal dysfunction in male Sprague Dawley rats**	Pharmacological prophylaxis and treatment with MTK	Ameliorated renal toxicity; ↓ responsiveness to acetylcholine; ↓ serum creatinine, blood urea nitrogen, LDH; ↑ serum albumin to normal levels; ↑ GSH levels; ↓ SOD levels	[220]
**Pyelonephritis induced by *Escherichia coli* in Wistar rats**	Pharmacological treatment with MTK	↓ severity of kidney damage and renal scarring; ↓ serum TNF-α, creatinine, blood urea nitrogen, MDA levels; ↑ GSH levels	[221]

According to Table 2, MTK is a very promising drug that shows a lot of potential in various disorders. Nowadays, the strategy of drug repurposing is being followed by many pharmaceutical companies [222] since it allows a faster and less cost-effective drug development, with the skipping of some development steps and clinical trial phases. In fact, some of the more known instances of drug repurposing and interest in off-target drug effects were the use of drugs in the treatment of COVID-19 symptoms during the pandemic, which included MTK [213]. Additional examples are the repurposing of minoxidil (an antihypertensive agent) as hair growth stimulant, and the use of sildenafil (developed for angina) in erectile dysfunction or pulmonary hypertension [222].

As MTK has been on the market since 1998, some of its proposed repurposing applications were based on retrospective studies, whereas others resulted from ADRs or serendipity [188,192,203,204]. Considering all the findings until now (clinical and in vitro/in vivo), it is critical to unveil the mechanisms involved in these new potential applications, in order to validate the repurposing uses. Once validated, patients could benefit from a potential new drug that will contribute to a better management of their symptoms, improving their quality of life. At present, most applications are based on the wide anti-inflammatory properties of MTK.

## 4. Human Neurodegenerative Diseases

Neurodegenerative disorders affect millions of people around the world and show increasing prevalence. These disorders are caused by the progressive degeneration and/or loss of a specific neuron population due to chronic neuroinflammation. The most common disorders are amyloidoses, tauopathies, α-synucleinopathies, and transactivation response DNA binding protein 43 (TDP-43) proteinopathies, which may cause movement or functional problems, such as Alzheimer’s disease (AD), Parkinson’s disease (PD), Lewy body disorders, and amyotrophic lateral sclerosis, among others [223].

Autopsies of older people have shown that aged brains develop abnormal accumulation of hyperphosphorylated Tau protein, amyloid-β deposits, accumulation of TDP-43, and α-synuclein deposits. Brains from people aged 90 and older have lost around 11% of their weight (approximately 150 g of brain tissue) when compared with people in their fifties. This weight decrease could be related to the loss of neurons and glia cells, myelin, fluid, or other factors. Despite the observation of these physiological changes in the aged brain, not all autopsied people suffered from neurodegenerative diseases, which suggests that some people might have compensatory mechanisms that enable them to maintain normal cognition [224,225]. During normal (healthy) ageing, intact synapses are maintained by APP processing through a non-amyloidogenic pathway, with amyloid production being balanced by clearance processes. The APP protein, as well as products resulting from its processing, play an important role in functions such as synaptogenesis, axonal growth, synaptic plasticity, learning, and memory. Furthermore, the Tau protein is involved in neuronal microtubule stabilization [226].

This section will further focus on two neurodegenerative diseases: Alzheimer’s and Parkinson’s diseases.

### 4.1. Alzheimer’s Disease

Nowadays, 47 million people suffer from dementia in the world, and it is expected that this number will increase to 131 million patients by 2050 [227]. Among these, around 80% of dementia is caused by Alzheimer’s disease [227,228]. AD is a progressive disease whose pathological changes start decades before the clinical symptoms, leading to the development of cognitive impairment, functional symptoms, and, later, dementia [227,228].

AD is classified in two forms: the familial early-onset (FAD) form and the sporadic late-onset (SAD) form. The FAD form affects approximately 5% of AD patients and is diagnosed in individuals between 30 and 60 years of age [67]. These patients present hereditary mutations in several genes involved in Aβ formations, such as the genes encoding β amyloid precursor protein (APP) and presenilin 1 and/or 2, which contribute to the early onset of symptoms [67]. The SAD form is mostly associated with age (patients older than 65 years of age), and the risk factors include pathways involved in cholesterol metabolism (APOE, CLU, and ABCA7), immune response (CR1, CD33, and Trem2), and endocytosis (PICALM and EPHA1, and ED2AP) [67,229,230,231,232,233].

AD is characterized by two major pathological hallmarks, namely the accumulation of β-amyloid (Aβ) plaques outside neurons and the accumulation of phosphorylated Tau protein, also known as neurofibrillary tangles (NFT), inside and outside neurons due to hyperphosphorylated Tau protein aggregation. The instability and reduced axonal transport caused by the loss of Tau function, as well as the formation of amyloid plaques, leads to damage and disruption of neuronal synapses and, later, cell death [234] (Figure 6B). Additionally, AD patients also display loss of neurons and white matter (brain atrophy and neurodegeneration), cerebral amyloid angiopathy (accumulation of amyloid plaques in the leptomeninges and small/medium-sized cerebral blood vessels, leading to fragile vessels), neuroinflammation, oxidative damage, and neurotransmitter imbalance [67,235,236].

The Aβ peptide is expressed as part of a 695-amino-acid polypeptide, the amyloid-β precursor protein, which is a glycosylated transmembrane protein encoded by a gene located on human chromosome 21 [237,238]. APP can be processed through two alternative pathways (Figure 6C): a non-amyloidogenic pathway, where APP is firstly cleaved by α-secretase, and an amyloidogenic pathway where APP cleavage is performed by β-secretase.

In the non-amyloidogenic pathway, α-secretase catalyses the release of a soluble amyloid precursor protein α (sAPPα) and an α-C-terminal fragment (CTFα or C83); the latter is converted by γ-secretase into an extracellular P3 peptide and an APP intracellular domain (AICD) peptide [239,240].

In the amyloidogenic pathway, β-secretase catalyses the formation of a β-C-terminal fragment (CTFβ or C99) and an N-terminal sAPPβ fragment. CTFβ is then cleaved by γ-secretase, releasing extracellular Aβ peptides of different length and the APP intracellular domain [239,240]. The most frequent final Aβ forms are the 40- (Aβ_40_) and 42-amino acid (Aβ_42_) peptides [240]. Aβ_42_ is more neurotoxic than Aβ_40_, possibly due to its higher tendency to produce oligomers [241].

**Figure 6 pharmaceuticals-15-01039-f006:**
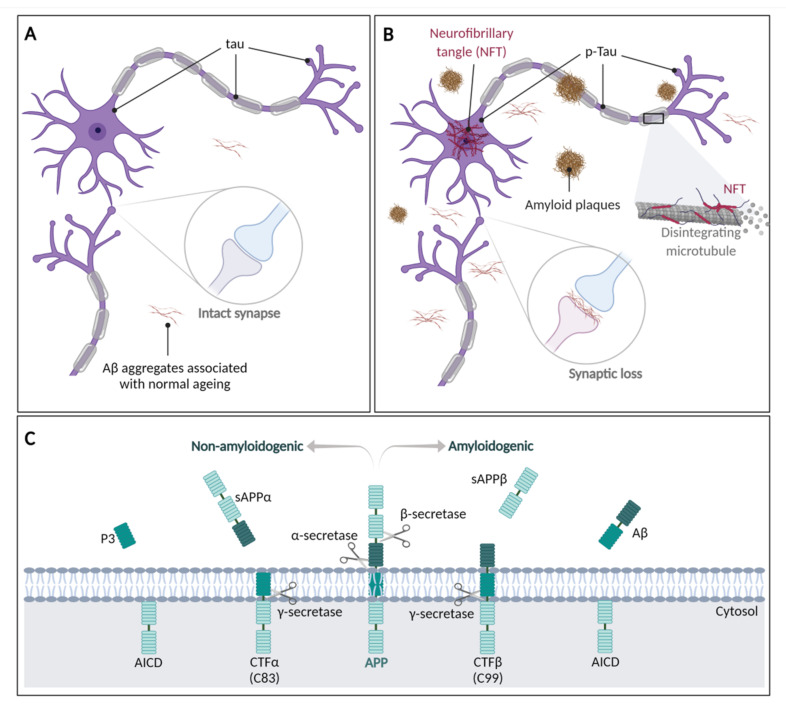
**The molecular basis of Alzheimer’s disease**. (**A**) Healthy cognitive ageing. (**B**) Alzheimer’s disease pathology. (**C**) Human APP proteolytic pathways. Adapted from Müller et al. [226], De Strooper et al. [239], Chen et al. [240]. Created with BioRender.com.

The accumulation of Aβ is caused by an imbalance between production, clearance (or degradation), and aggregation of peptides [236]. Whereas mutations in *APP*, *Psen1* and/or *Psen2* genes are associated with Aβ processing in FAD [242], SAD is linked to an accumulation of Aβ caused by a decrease in clearance mechanisms [243]. Clearance can occur within the brain or after transport from the brain to the periphery (liver and kidney) and includes proteolytic pathways that depend on neprilysin (NE), insulin-degrading enzyme (IDE), matrix metalloproteinases (MMPs), angiotensin-converting enzyme (ACE), endothelin-converting enzyme (ECE), plasmin, the activity of the ubiquitin–proteosome system, the autophagy–lysosome system, or microglial phagocytosis [243,244].

In addition to the characteristic Aβ peptide plaques and to phosphorylated Tau accumulation, the brain of some AD patients contains high levels of α-synuclein. This protein is expressed in kidney, blood cells, and, predominantly, in neurons; it is mainly known for its association with Lewy bodies and Parkinson’s disease pathologies. When present in high levels, α-synuclein forms oligomers and fibrils. Moreover, recent studies have suggested that amyloid plaques are able to promote the formation of α-synuclein aggregates, increasing neurotoxicity [245,246].

Currently, no treatments are available for Alzheimer’s disease. However, some drugs that temporarily improve disease symptoms are currently used: rivastigmine, galantamine, and donepezil, which are cholinesterase inhibitors, increasing neurotransmitter levels in the brain; and memantine, an antagonist of the *N*-methyl-D-aspartate receptor. Recently, aducanumab, a monoclonal antibody targeting the *N*-terminal pyroglutamate Aβ epitope, which could help in the reduction of the amyloid plaque level, was the first drug of this class to be approved by the US FDA [240,247,248].

### 4.2. Parkinson’s Disease

According to the British National Health Service (NHS), 1 in 500 people suffer from Parkinson’s disease. This neurodegenerative disorder is mainly characterized by the loss of dopaminergic neurons in the midbrain substantia nigra pars compacta region, which causes motor and nonmotor symptoms. The motor symptoms, collectively known as parkinsonism, include bradycardia, rigidity, resting tremor, and impairment of postural balance. Nonmotor symptoms include depression, anxiety, sleep disturbance, constipation, dementia, and cognitive decline [89,249,250].

As mentioned, the hallmark of PD is the progressive loss of dopaminergic neurons present in the substantia nigra, with the appearance of Lewy neurites and Lewy bodies—intracellular inclusions of α-synuclein aggregates—that ultimately lead to dopaminergic neuron death [249,250]. In addition to α-synuclein, Lewy bodies can also contain misfolded phosphorylated Tau and Aβ proteins, increasing neuronal toxicity [250]. Thus, the mechanisms involved in Parkinson’s disease pathology include the accumulation of misfolded protein aggregates, loss of protein clearance mechanisms, mitochondrial damage, oxidative stress, excitotoxicity, and neuroinflammation [250].

Besides dopaminergic neurons, serotoninergic neurons are also involved in PD. Dysfunctions in serotoninergic neurotransmission contribute to motor and nonmotor symptoms, such as resting tremor, dyskinesia, depression, and anxiety [251,252].

Currently, no treatment is available for Parkinson’s disease and the prescribed drugs only allow the control of some symptoms: levodopa (a precursor of dopamine), dopamine receptor agonists, inhibitors of monoamine oxidase B (MAO-B), catechol-*O*-methyltransferase (COMT) inhibitors, amantadine (an anti-influenza drug widely used in parkinsonism and dyskinesia), and anticholinergic drugs, including antidepressant drugs [89]. The use of MAO-B inhibitors intends not only to decrease the metabolism of neurotransmitters such as dopamine, increasing their extracellular concentration, but also to reduce the oxidative stress produced by MAO-B activity [253].

MTK repurposing for AD and PD management has been investigated in several works, using various biological systems, from cell-based systems to transgenic animals, as reviewed in Table 2. Most of the published studies indicate that MTK administration leads to a diminished inflammatory status at the CNS level, in agreement with MTK actions as CysLTR_1_ antagonist and with the contribution of neuroinflammatory glial processes to neurodegeneration progress. However, and until the mechanisms underlying MTK’s toxicity are fully elucidated, any potential applications of MTK beyond the officially approved ones must be considered with extreme care.

## 5. Conclusions

MTK is a widely used leukotriene antagonist drug, targeted at asthma management for adults and children alike. Regarding lactation, MTK levels found in human milk were below therapeutic ranges established for children, supporting the safety of this treatment for asthmatic breastfeeding mothers [254].

MTK’s anti-inflammatory action is not confined to the respiratory system but is of a more systemic nature, which has led to the development of clinical studies aiming at MTK repurposing for various inflammatory-based conditions, particularly aiming at the management of a number of neurodegenerative diseases.

However, MTK is associated with a number of adverse drug reactions, particularly at the CNS level, where neuropsychiatric events are linked to MTK administration but promptly resolve upon stopping the treatment. Although the molecular basis for these toxic side effects is still unknown, they must be taken into consideration when addressing the potential repurposing of MTK.

The metabolism of MTK has been studied extensively by various authors, as reviewed above. A link between metabolism and toxicity was not identified from the various observed phase I and phase II metabolites, indicating that MTK’s toxicity is likely mediated via interaction with biological pathways and not through chemical reaction with biomolecules.

## Figures and Tables

**Figure 1 pharmaceuticals-15-01039-f001:**
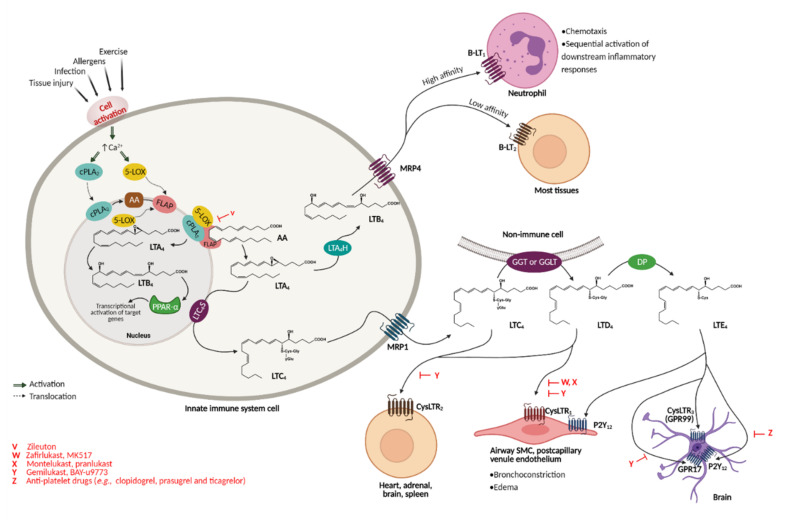
Leukotriene biosynthesis pathway and receptor recognition. Leukotrienes are synthesised upon activation of the immune system through an LT biosynthesis cascade, acting on various organs through different receptors. V is a 5-LOX inhibitor, whereas W, X, Y, and Z are inhibitors of the leukotriene pathways; all are marked in red. Image adapted from Funk [19]. Created with BioRender.com.

**Figure 3 pharmaceuticals-15-01039-f003:**
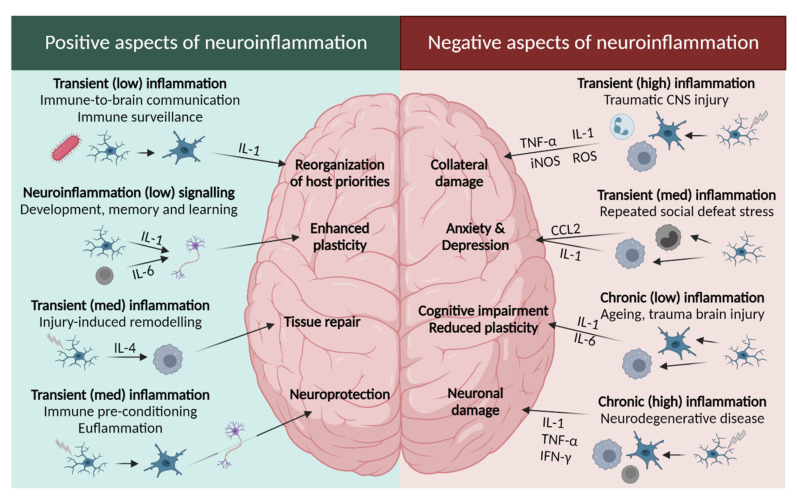
**Positive and negative aspects of neuroinflammation**. The consequences of neuroinflammation depend on the duration and severity of the immune response. (**Left**) induction of sickness behaviour to restore the host’s homeostasis after infection; (**right**) chronic neuroinflammation tends to carry negative consequences. Low, med, and high refer to a low, medium, and high levels of inflammation. Image adapted from DiSabato et al. [68], with permission from John Wiley and Sons. Created with BioRender.com.

**Figure 4 pharmaceuticals-15-01039-f004:**
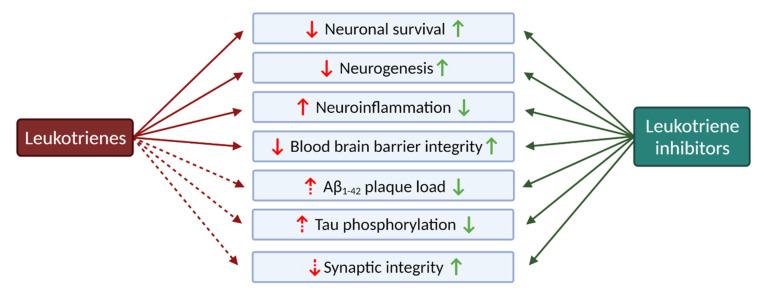
Leukotrienes in the central nervous system: a pleiotropic effect. Broken lines symbolize putative actions of leukotriene signalling that require further investigations. Image adapted from Michael et al. [67], with permission from Elsevier. Created with BioRender.com.

**Table 1 pharmaceuticals-15-01039-t001:** Contribution of different CYP isoforms to MTK metabolism in human liver microsomes. The colour gradient indicates the relative contribution of each CYP isoform: the darker the shade, the more relevant is the CYP isoform role.

			Recombinant CYP Isoform											
			1A2		2A6	2B6	2C8	2C9	2C19	2D6	2E1	3A4	3A5	
M2a														[121]
														[123]
														[124] *
														[122] *
M2b														[121]
														[123]
M3														[121]
														[123]
														[124]
M4														[123]
M5a														[121]
														[123]
														[124]
														[122] ^δ^
M5b														[121]
														[123]
														[124]
M6														[121]
														[123]
														[124]
														[122]
					Metabolite formation rate (no shade—lower; darker shade—higher)									
	Not included in the study													

* No distinction between M2a and M2b diastereomers was made. ^δ^ No difference between M5a and M5b was reported.

## Data Availability

Not applicable.

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
