# Peer review of "Leukotrienes vs. Montelukast—Activity, Metabolism, and Toxicity Hints for Repurposing"

_pharmaceuticals, 2022, doi:10.3390/ph15091039_

Round 1
Reviewer 1 Report
The review is comprehensive and addresses an important topic. However, there are several formatting issues and further this reviewer felt that the authors should elaborate more in the discussion about what they propose would be important topics to further the field.
Formatting issues:
1) Several citations show as an error. This should have been fixed prior to review.
2) Spelling errors in the figures (e.g., Figure 2 "impaiment" should be "impairment"
3) Strange line break formatting e.g., lines 232 & 239.
4) dead links in lines 260 & 277, unclear what it is supposed to be pointing to.
Reviewer 2 Report
The review publication submitted for peer review concerns Montelukast and its new applications due to its interaction with leuktriene receptors and other targets. The authors have demonstrated a fluent knowledge of the subject and have comprehensively described, based on almost 250 literature references, both the mechanism of action of MTK itself, its side effects and new developments. The review is well organized and written in clear language, making it a very good read even by the standards of a scientific paper. However, I found several issues that should be corrected before its publication:
1. Lines 75-78, it would be of great use to show the structures of the mentioned drugs, along with some biological activity data (eg. to CysLT receptors)
2.The Figure captions are definitely too large – please move the descriptive parts to the manuscript body
3. Figure 2, sentence “Positive and negative aspects of neuroinflammation.” Please delete this sentence, as it appears (in different way) in the second sentence
4. Lines 205-207, what exactly is the evidence that supports the findings? Please elaborate a little more on this.
5. Lines 260 and 277 “described in more detail in 0” unclear – I assume the reference to end paragraphs is missing
6. Line 301 – there is an inconsistency with the chemical name of MTK: name is 7-chloro… and the structure on the Figure 4 states that the chlorine atom is in position 8, which is incorrect. Please check the heteroaromatics numbering guidelines.
7. Lines 499-501 can be moved to “Conclusions” part
8. Table 2 is somehow hard to read, due to missing Applications resulted from division of the tables onto different pages. Moreover, in my opinion Paragraph 3.3. which seems to be key paragraph of this review is lacking the descriptive information – most of it is in the Table (which footer, btw. Should be shortened/moved to abbreviates section). It would be good to at least describe the key repurposing directions in a little more detail. Please elaborate more, or discuss the results from the table.
9. Figure 5 – similarly, please move the description to the manuscript body.
Along with some editorial errors:
10. In some places the Authors forget to use italics, eg. : via, in vitro etc..
11. Figure 1 – the chemical structures are hardly readable, please correct
12. The Table/Figure references are missing, or simply replaced by “Error! Reference source not found.” due to pdf enclosure from docx file most probably. Try “print to pdf" option instead of "save as"
13. the word splitting to the next line also fails (eg line 43, 239 etc)
